# AI, Robot Neuroscientist:
# Reimagining Hypothesis Generation

**Jiaqi Shang**[‡]
Program in Neuroscience
Harvard Medical School
Boston, MA 02115
jiaqishang@g.harvard.edu

**Will Xiao**[‡]
Department of Neurobiology
Harvard Medical School
Boston, MA 02115
xiaow@g.harvard.edu

## Abstract

Neuroscience has long relied on human-conceived hypotheses, yet the brain's complexity fundamentally challenges this epistemology. Modern technologies and the large-scale data collection they enable throw this challenge into sharp relief. We champion the potential of AI for neuroscience exploration. We highlight both implicit, 'uninterpretable' models as aids in hypothesis formulation and symbolic regression for explicit hypothesis generation. For researchers from non-neuroscience backgrounds, we discuss domain-specific considerations in integrating AI into neuroscience research. By spotlighting the underexplored avenues for AI to accelerate neuroscience, we aim to induce both communities toward these exciting research opportunities.

## 1 Introduction

The brain is a complex matter, about which it is challenging to formulate hypotheses. Hypothesis generation in neuroscience has depended on heuristics that range from adapting psychology concepts [1], to simplified neuron [2, 3] and network models [4], anatomical localization, and serendipity [5].

Meanwhile, neuroscience is data-rich—current technology allows for simultaneously recording tens of thousands of neurons, a number that continues to grow in a Moore-like law [6]. Current analyses have just begun to make sense of these data, often with linear methods [7, 8, 9]. While linear decomposition can already detect on the order of $10^2$ dimensions in the activity of $10^4$ neurons, the interpretation of these dimensions is often further restricted to only a handful of task variables [10, 9]. Finally, much of high-dimensional neural activity is ascribed to 'mixed selectivity' that resists unraveling into simpler principles [11, 12].

The combination of complexity and data abundance makes neuroscience especially fertile ground for AI-driven hypothesis discovery. AI methods are uniquely suited to detect intricate patterns hidden in neural data. AI research has developed a plethora of tools suited for specific neuroscience questions. Much of this potential remains untapped, as neuroscience today still applies AI tools sparsely. Here, we review current statistical and AI methods for analyzing neural data and identify promising directions for future work.

### 1.1 Related perspectives

Many recent perspectives highlighted the potential for AI to inform neuroscience [13, 14, 15]. These perspectives propose AI models as embodiments of pre-existing hypotheses to be verified on neuroscience data. Examples of such hypotheses include temporal-difference learning, external

---

[‡]Equal contribution.

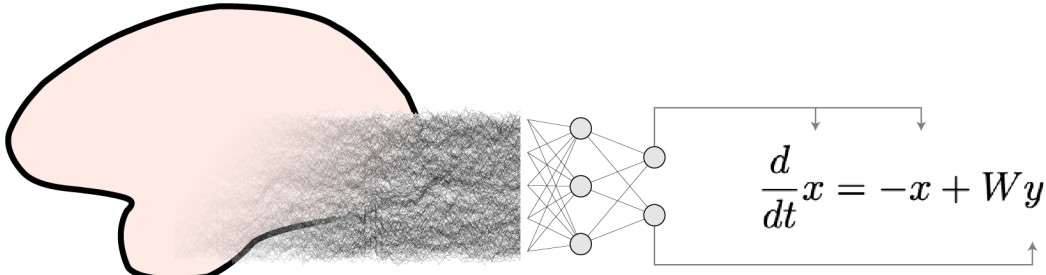

Figure 1: AI can help model and discover hypotheses from raw neural data.

memory, meta-reinforcement learning [13], task-optimized convolutional neural networks (CNNs) [16], and the learning algorithm, which is further analyzed into three components: the learning rule, objective function, and architecture [14]. In other words, these perspectives advocate for a systems identification approach, exemplified in work such as [17]. Comparing models and the brain to infer shared design principles has yielded some insights and also met challenges [17, 18].

We put a different emphasis. Instead of building hypotheses into AI and interpreting the model-brain match in those terms, we advocate for AI as a tool for hypothesis discovery from neural data (Figure 1). Our perspective is closer in spirit to [19], while we focus on the potential for AI to deliver conceptual neuroscience insights. Although we briefly discuss the potential of AI methods for neural data preprocessing, we prioritize the analysis of functional neural dynamics during perception, cognition, and behavior. Finally, we focus on large-scale electrical and optical physiology data because they best represent the rich, complex patterns that AI thrives on while promising further scaling.

## 2    Current methods for analyzing large neural data

### 2.1    Dimensionality reduction

Dimensionality reduction techniques are commonly used to analyze high-dimensional neural data. These methods transform the data into a lower-dimensional space while preserving the inherent variance. For instance, Principal Component Analysis (PCA), when applied to neural data from the mouse visual cortex [7], reveals a significant correlation between the first principal component and animal arousal indicators such as running, whisking, and pupil dilation. Other techniques, including Factor Analysis (FA), t-distributed Stochastic Neighbor Embedding (t-SNE), and Independent Component Analysis (ICA) are also frequently employed [20, 21, 22]. After the reduced dimensions are extracted, they can be regressed against observable experimental variables to infer underlying neural mechanisms. A strong correlation between an activity dimension and its predictions from certain variables indicates the dimension contains information about those variables.

### 2.2    Dynamical models

A notable limitation of methods like PCA is that they treat each time point as independent. Temporally patterned behaviors, such as reaching, likely involve neural mechanisms with a temporal structure. Several methods have been proposed to explicitly address the temporal structure in neural mechanisms. They model the temporal evolution of the neural dynamics using tools such as Gaussian processes [23, 24], dynamical systems [25, 26, 27, 28, 29] and Recurrent Neural Networks (RNNs) [30, 31]. For example, Latent Factor Analysis via Dynamical Systems (LFADS) [30] uses an RNN to reconstruct recorded data across trials. After optimizing, the RNN can accurately predict behavioral variables, such as reaching directions for previously unseen neural data.

### 2.3    Latent variable models

Neural mechanisms often encompass factors not directly observable. Many complex cognitive processes involve multiple stages of input processing before behavioral outputs. To uncover these hidden factors, researchers have turned to Autoencoders, a type of artificial neural network (ANN). Autoencoders compress neural data through an encoder to produce a latent representation; this

representation is then used by a decoder to reconstruct the original data. This latent representation thus captures hidden features reflective of the intrinsic neural mechanisms. To illustrate, pi-VAE [32] employs the variational autoencoder to analyze hippocampal recordings from mice during a spatial navigation task. The model effectively extracts latent factors that separate the spatial and temporal information related to navigation. While Autoencoders can capture intricate mappings from neural data to latent representations, challenges persist in interpreting the extracted latent representation. Notably, interpretation is often confined to preconceived hypotheses, such as correlating with sensory inputs or motor outputs. Such a constraint can limit the breadth of hypothesis exploration, potentially missing out on novel neural mechanisms that operate beyond these predefined parameters.

## 2.4 Encoding models for visual neurons

Encoding models that allow image-computable predictions of visual neuron responses are among the first applications of AI in neuroscience [33, 34] and remain an active research direction. These models comprise a nonlinear core that is often trained without neural data and a linear mapping that is fitted to transform the model-core output to neural activity. Model cores trained without neural data thus do not extract any latent neural activity structure, although models that learn end-to-end from neural data pooled across many animals may represent implicit structures. Encoding models have been used, as advocated in related perspectives (Section 1.1), to instantiate preconceived conceptual hypotheses such as the convolution motif, learning objective and rule [14], and topological cortical organization [35, 36, 37].

## 2.5 Needs unmet by current methods

Current methods focus on discovering latent factors that have predictive power within a given dataset. The latent factors are sometimes given interpretations by correlation to external observables like stimuli and behaviors. However, these methods do not introduce novel concepts or rules that can extrapolate across datasets. In contrast, any scientific theories a 'robot neuroscientist' can produce should ultimately describe properties of the system (not merely of the data) that extrapolate to a wide range of situations unseen during model fitting [19].

# 3 'Uninterpretable' models as aids in hypothesis formulation

Connectionist AI models are opaque by default: An active research field of explainable AI is devoted to interpreting models. However, even 'uninterpretable' models, when coupled with an appropriate scientific framework, can help scientists form hypotheses. Here we use 'uninterpretable' to refer to models that are not expressly designed to be interpretable. Even though AI models are mathematically well-defined and explainable in this sense, such models do not explicitly represent conceptual insights. Below, we discuss two ways in which uninterpretable models can aid scientific discovery: feature attribution and factorizing complex interdependencies in data.

## 3.1 Feature attribution

Feature attribution on performance-optimized models can reveal relations between features in the data, thereby helping human scientists form hypotheses (Figure 2, left). Davies et al. (2021) used this approach to help mathematicians formulate two conjectures, one in knot theory and the other in representation theory, which were subsequently proven manually. Take the knot theory case for example. Davies and colleagues searched for unknown relationships between the *geometric invariants* and *algebraic invariants* of a knot by training supervised ANNs to predict the latter from the former using a synthetic dataset. They found that a knot $K$'s signature $\sigma(K)$, an algebraic invariant, can be predicted above-chance from the geometric invariants of $K$ and used feature attribution to identify the top three contributors: the meridional translation $\mu$ (real and imaginary parts) and the longitudinal translation $\lambda$. Human mathematicians subsequently conjectured and proved a novel theorem involving these quantities. (Notably, other geometric invariants entered the final theorem but had much lower attribution scores than the top three.) In the representation theory example, neural networks helped identify features of the Bruhat interval, a large directed graph too cumbersome for easy intuition, that ultimately led human mathematicians to a theorem and its (conjectured) generalization.

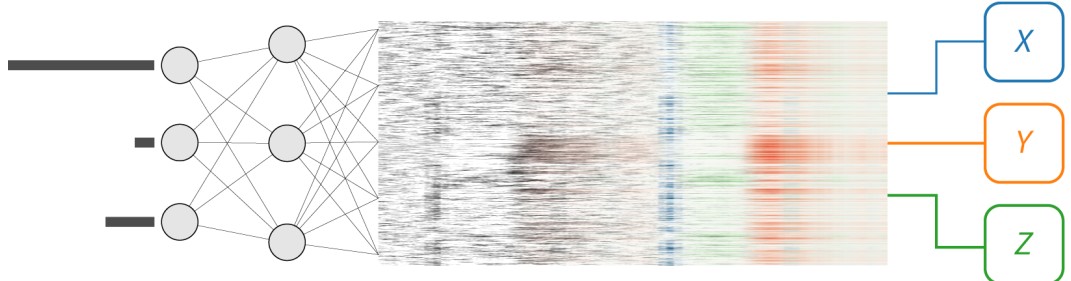

Figure 2: AI models can help evaluate feature contributions (left) to neural activity (center) or represent factors in the data (right).

In the feature attribution approach we propose, the AI model serves an analogous function to a generalized linear model (GLM). Another analog, symbolic regression [38, 39], promises to directly infer mathematical formulae that describe the laws governing a system; we discuss it in Section 4. However, deep learning with feature attribution enjoys some advantages over GLMs and symbolic regression. First, deep learning can learn complex, non-linear relations that are hard to capture in a GLM or a prespecified functional basis. Second, the same core architecture, such as the Transformer, can perform well across data domains [40, 41, 42]. Thus, an AI-based approach may be easier to apply across scientific fields.

In neuroscience, feature attribution can help identify the task and stimulus variables that most contribute to the ubiquitous and diverse mixed selectivity in neurons [11, 43]. Investigators can further use 'ablations'—withholding certain variables—to narrow the sufficient set of variables.

## 3.2  Factorizing complex dependencies in data

The ability of AI models to capture subtle interdependencies in the data can be leveraged to factor out, or control for, correlated variables in imbalanced data and allow scientists to ask otherwise intractable questions (Figure 2, right). For example, an intriguing study [44] examined whether non-robust image features (those alterable with small perturbations to the image) contain useful information for classification. It is not trivial to define robust and non-robust features in closed form, let alone manipulate them in high-dimensional images. However, non-robust CNNs implicitly represent non-robust features, and vice versa for adversarially trained CNNs. Thus, using the two types of CNNs and gradient descent, Ilyas and colleagues could create images that mostly contained robust or non-robust features for training classifier networks. They also trained networks to classify adversarial images according to *target* labels, because these images can now be interpreted as containing non-robust features that conflict with the remaining features. (E.g., the images look 'wrongly' labeled to humans.) Classifiers trained on either the non-robust or the conflicting features have good accuracy on unmodified validation images, indicating that non-robust features indeed contain useful information for categorization. This study exemplifies how AI models allow scientists to investigate the concepts models implicitly represent.

There is increasing interest in neuroscience to explain naturalistic data. However, drawing reliable conclusions from natural data is hindered by potential spurious correlations among variables [45]. AI methods can better 'regress out' uninteresting variables than linear regression and decomposition techniques. For example, the mouse visual cortex is strongly modulated by dimensions that are linearly independent of the visual stimulus but correlated with non-visual variables like pupil size, face movements, and running speed [7], as discussed in Section 2.1. However, some of the non-visual activity dimensions may be nonlinearly explainable by the visual activity, AI models can more fully remove the visual contributions to examine non-visual responses.

## 3.3  Limitations

Despite ways to derive insights from uninterpreted AI models, this process still requires a scientist in the loop to prespecify variables of interest. Ultimately, we desire an AI neuroscientist to extract 'concepts'—e.g., sparse features, relations between features, and connections to existing scientific and mathematical concepts. The next section reviews AI methods that hold such promises.

# 4 Methods that generate interpretable hypotheses

## 4.1 Symbolic regression

Recent advances in AI, especially in symbolic regression, offer promising avenues for discovering interpretable neural mechanisms from data. Symbolic regression aims to capture structure in the data with mathematical equations, bypassing the constraints of traditional methods where the structure, such as the linear subspaces in PCA, has to be presumed. The mathematical equations are constructed from a set of basic operators (such as addition) and basis functions (such as polynomials and trigonometric functions), and an optimization procedure selects the expression that both is succinct and fits the data well. As an example, Schmidt and Lipson [38] demonstrated that given recorded time-series data of position, velocity, and acceleration of a two-spring single mass system, symbolic regression can reproduce Newton's second law.

Though the optimization challenges in symbolic regression are substantial, recent advancements in heuristic methods using techniques like genetic programming [38, 46, 47, 48], Bayesian methods [49], sparse regression [39, 50] and neural networks [51, 52, 53] provide effective solutions. Furthermore, the development of specialized toolkits [54] has bolstered the applicability of these methods. Given these advancements, symbolic regression is poised to offer groundbreaking insights into neuroscience.

## 4.2 Applying symbolic regression to neuroscience

How can symbolic regression aid in deciphering neural mechanisms from data? We explore this with a specific example. Neural responses to an identical stimulus have been empirically observed to differ across trials [55, 56, 57] and this variability is correlated across neurons [58, 59]. However, the mechanisms driving this neural variability remain undefined.

Traditional neural analysis methods, reviewed in Section 2, have yet to fully explain this variability. Neuroscientists have approached this problem by positing models based on various hypotheses. These hypotheses explored how a single factor of population activity could modulate the neural stimulus-response, either through additive offset [60, 61], multiplicative gain [62], or a hybrid (termed 'multi-gain') [63, 64, 65]. Comparing the fit of each model to experimental data indicated a predilection for the multi-gain model. This model-fitting approach has limitations: it rests on predefined assumptions about the underlying mechanisms. It is unclear whether the assumptions hold and conceivable that the neural sensory response is modulated by population activity through alternative nonlinear interactions.

Symbolic regression emerges as a potent method capable of unveiling mathematical relationships in data without relying on pre-defined models (Figure 3). To understand the link between population activity and neural sensory responses, we can create an input-output pairing dataset. Inputs include sensory stimuli (e.g., orientation for drifting gratings) and neural population activity, while outputs capture neuronal activity in the visual cortex. Following Udrescy and Tegmark [52], we first fit the dataset to a black-box model, such as a multi-layer perceptron, to capture the input-output relation. This model is geared to accurately predict unseen neural activity and offers a more generalizable data

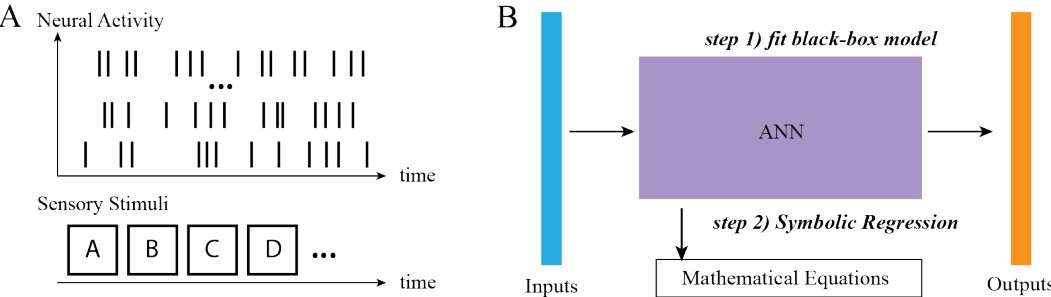

Figure 3: Applying symbolic regression to neuroscience. A) Raw data, including neural activity and sensory stimuli. B) The workflow involves structuring the raw data into input-output pairs, fitting with a black-box model, and applying symbolic regression to extract mathematical equations.

representation, facilitating the exploration of data properties, especially at input values absent from the dataset. Using symbolic regression, we then distill mathematical equations from this model, revealing an experimentally testable formula for how population activity affects neural sensory responses.

If the resultant equation features combined multiplicative and additive terms, it suggests that the population activity has a multi-gain effect, aligning with conclusions from previous model-based approaches. Nevertheless, the modulatory effect could potentially be more intricate. Symbolic regression enables an exploration of the equation landscape more expansive than the previous manual model formulation approach. Consequently, symbolic regression holds the promise of unveiling novel neural mechanisms in a less biased and more efficient manner.

### 4.3  Limitations: identifying variables

While symbolic regression offers a promising approach for discovering mathematical relations within data, its success heavily relies on the appropriate choice of input variables. As an example, Schmidt and Lipson [38] showed that given only the position and velocity data for a two-spring system, symbolic regression converged on the energy laws instead of Newton's second law. The need to select the right variables poses a challenge: the brain is vastly complicated, spanning multiple levels of complexity and making it difficult to preemptively identify the most pertinent variables, which are liable to be complex functions of the raw neural activities. Future work should ideally explore how to extract both the relevant variables and their mathematical relations from neural data.

### 4.4  Other methods

Discovering both variables and their relations within data is at the frontier of AI research. This problem is challenging due to the expansive search space and likely requires proper inductive biases [66]. Nevertheless, recent advances offer promise. For example, Chen et al. [67] introduced a method to identify state variables in high-dimensional data. Given an observed system, the algorithm first models the system's dynamics using a black-box neural network with bottleneck latent embeddings. It then estimates the number of state variables from these latent embeddings using geometric manifold learning techniques. Lastly, the algorithm trains another latent reconstruction neural network with the exact identified number of latent variables to identify the system's governing mechanisms. The algorithm was demonstrated to successfully extract the angle and angular velocity as key variables from a video of a pendulum in motion. Such techniques hold exciting potential for unveiling complex patterns and structures within neural data.

Casual discovery [68, 69, 70] is another promising avenue for progress. Here, the aim is to learn the ground-truth causal generative process of the data. For instance, Shen et al., 2020 [71] explored an enhancement to the VAE (Section 2), where the latent factors can be causally related. One advantage of integrating causality is its capacity to support intervention on causal variables. Such intervention predictions can then be validated experimentally through neural perturbation studies [72].

## 5  Neuroscience-specific considerations

### 5.1  Finding the right level of abstraction for hypotheses

Unlike the data AI typically handles, such as images or text, the brain is a complicated system spanning multiple levels of complexity: molecules, including neurotransmitters and receptors. regulate processes like synaptic transmission; individual neurons integrate inputs and relay signals to their neighbors; assemblies of neurons form microcircuits dedicated to specific computations; finally, groups of microcircuits converge to execute cognitive functions. Crucially, these levels are closely linked: Changes at one level can propagate and cause changes at another level. For example, molecular changes can alter neuron functions and even affect behavior [73, 74].

The brain's deep hierarchical structure presents distinct challenges when employing AI to elucidate neural mechanisms. When AI proposes a neural 'law,' it is crucial to interpret it in the appropriate biological context. For example, mechanisms identified by AI at network levels may not decompose neatly into cellular or molecular mechanisms, thereby missing a mechanistic level of explanation. Conversely, some key mechanisms may only emerge when examining data across multiple levels. One way to navigate these complexities is to integrate existing knowledge in neurobiology. For

instance, preconfiguring AI models with a set of biologically plausible canonical computations can enhance interpretability and relevance [75].

## 5.2 Non-stationarity

Biological systems are adaptive and multi-scaled, properties that pose additional challenges to AI-for-science techniques developed for physical and engineering sciences. For example, symbolic regression to discover physical laws assumes that the same equation describes system evolution in a time-invariant way. However, the brain is plastic at multiple time scales. We provide a representative but non-exhaustive list to illustrate the relevance of non-stationarities. At the $10^{-2}$–$10^0$ s scale, neural activities evolve due to circuit properties like adaptation and recurrent processing even without changes in the external drive (e.g., viewing a static image). At the $10^0$–$10^2$ s scales, sensory responses continue to manifest adaptation, while internal brain states including arousal, attention, motivation, and neuromodulation also fluctuate. At yet longer time scales (hours to days), the brain manifests the effects of learning. At time scales of a species' life history (birth to adulthood to senescence), the brain develops, remodels, ages, and evolves. Theseus' paradox involves all of these scales.

## 5.3 Data modalities

Neuron-level brain recording today remains in the regime of extremely sparse sampling. Leading edge techniques can record about $10^4$ neurons, but a tiny fraction in moderately complex brains. For comparison, the mouse brain contains on the order of $10^8$ neurons, while the human brain has almost $10^{11}$. Although recording from most neurons in a mammalian brain is ultimately possible physically [76], sampling by current recording techniques—primarily calcium imaging and high-density electrophysiology like Neuropixels—is sparse and biased, with the two techniques differently deviating from i.i.d. The neocortex is locally organized like a flat sheet with layers (imagine puff pastry but with six layers of varying depth). Neuropixel recordings are confined to a narrow sheath around a linear track (the voltage signal amplitude decreases ten-fold $\tilde{5}0$ um away from the neuron [77]; this slender track may be normal or tangential to the cortical surface. Meanwhile, calcium imaging more evenly samples the tangent plane but penetrates to a limited depth (typically layers 2/3) within accessible brain areas (e.g., at the top of the head and outside cortical convolutions).

Calcium imaging uses genetically encoded fluorescence reporters that restrict the signal to a defined cell type and, usually, a sub-cellular compartment (e.g., cell bodies vs. dendrites). In contrast, electrophysiology provides essentially no cell type or compartment information and is probably dominated by large neurons that generate high-amplitude spikes.

The time resolution also differs between the two modalities. Ephys has sub-spike resolution (below 1 ms), whereas calcium imaging is currently limited to about $10^{-1}$ s by the indicator dynamics [78].

Ephys comes with non-single-neuron signal types—multiunit activities (MUAs) and local field potentials (LFPs)—that have no close analogs in calcium imaging. MUAs are functionally and conceptually (under a linear readout assumption) similar to single-unit activity, whereas LFPs have distinct interpretations.

# 6 Other opportunities

## 6.1 Data preprocessing

Data preprocessing uses statistical methods that AI can supply. For clearly defined end goals (e.g., to localize cells, align imaging sessions, and sort spikes), current methods serve well and leave less margin for improvement by AI. However, AI may reveal additional information in raw data in hard-to-expect ways. Just as images contain category-relevant 'adversarial features' unrecognizable to humans, so there may be informative patterns amidst the diffuse fluorescence in calcium images after cell extraction or within the voltage trace fluctuations after spike sorting, although we stress that the existence of any residual information and its form are unexpectable by definition. Thus, data preprocessing may yet hold surprising rewards for applying AI.

## 6.2 Benchmark datasets

Large-scale, high-quality datasets are pivotal for the advancement and evaluation of AI methods. In neuroscience, there have been concerted efforts to offer standardized and accessible large-scale datasets. For instance, Brain-Score [18] assesses the feature alignment between task-pretrained deep neural networks and the primate ventral visual stream. The Allen Institute for Brain Science [79] and the sensorium competition [80] provide expansive data from mouse visual areas in response to natural images. The Neural Latent Benchmark [81] provides monkey responses across sensory and motor areas for a diverse set of cognitive behaviors.

The majority of these datasets are designed for predicting neural activity from sensory stimuli. While building precise prediction models is useful, interpreting those models is also vital. A comparable initiative in the realm of physics is the Feynman Symbolic Regression Database [52]. This database is organized into input-output pairs: the inputs are observational data, and the outputs correspond to equations of physical law. By this token, an aspirational dataset for neuroscience would have neural data as inputs and equations that describe those data as outputs. Yet, neuroscience poses a distinct challenge: The ground-truth mechanisms are often unknown. A pragmatic intermediary approach might entail representing outputs as AI-generated testable predictions. These predictions could then be either cross-referenced with existing literature or tested experimentally.

## 6.3 Using prior knowledge

Scientific discoveries do not occur in a vacuum [82]; they heavily depend on and build upon previous knowledge. In formulating hypotheses, prior knowledge not only constrains the hypotheses to ensure alignment with established knowledge but also sparks inspiration by providing frameworks for novel ideas. When generating neuroscience hypotheses using AI methods, it is crucial to incorporate prior knowledge. For example, in feature learning 3, established neuroscientific principles, such as the firing patterns of specific neural types, can be integrated using regularization terms to guide the learning process. Neuroscience knowledge presented as concepts or equation-based models can be directly incorporated into symbolic regression 4 as candidate variables or starting points, thereby guiding the symbolic search.

# 7 Conclusion

AI has begun to transform all fields of science [83, 84]. If any field should be exceptional, it is neuroscience: The study of natural information processing systems has always been tightly entwined with AI, in history and inherently. We reviewed current applications of AI in neuroscience, which emphasize their parallels. Further from the streetlight, we spotlight opportunities for using AI as analysis tools to unlock insights into neuroscience data.

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
